# $\alpha$Max-B-Cubed: A Supervised Metric for Addressing Completeness and Uncertainty in Cluster Evaluation

## Abstract

Assessing the quality of clustering results is a crucial and challenging task. The B-CUBED ($B^3$) precision and recall evaluation metric has gained popularity due to its ability to meet four formal constraints: homogeneity, completeness, rag bag, and size vs. quantity. However, the 'completeness' constraint, which demands that items of the same category be grouped in the same cluster, can pose problems for finer clustering algorithms that identify sub-clusters within clusters. This issue is particularly pronounced when the available labels are imprecise and coarse, resulting in uncertain and fuzzy cluster evaluations. To address this issue, we propose a modified evaluation metric called $\alpha$Max-$B^3$. Our approach accounts for completeness and uncertainty in subgroup evaluation by reorganizing clusters into super-sets based on the most prevalent label and evaluating them alongside the original clusters using a modified weighted $B^3$ metric. The extent of uncertainty, given by $1 - \alpha$, can be either explicitly specified or automatically estimated.

## 1 Introduction

The evaluation of clustering methods and their results is a complex task due to the lack of clear-cut criteria for determining the quality of clusters (Rand, 1971). While clustering may seem simple in theory, it is difficult to create a general framework that works for all cases (Rai & Singh, 2010; Xu & Tian, 2015; Berkhin, 2006; Xu & Wunsch, 2005). In fact, in "An Impossibility Theorem for Clustering" (Kleinberg, 2002)", an example of three simple properties was presented for which no clustering function could satisfy all three at the same time, exposing inevitable trade-offs. Follow-up work (Ben-David & Ackerman, 2008) proposed considering clustering quality measurements as the object to be axiomatized rather than clustering functions, and proposed a revised set of criteria (axioms) for such measures. The authors show that the *clustering-quality* framework is richer and more flexible than *clustering functions* because it allows the postulation of axioms that capture the features expressed by Kleinberg's axioms without producing a contradiction. The evaluation of clustering methods is therefore important due to the difficulty in developing a unified clustering framework that is independent of any underlying algorithm, objective, or model. Approaches to formalizing such qualitative objective criteria are mainly distinguished between two categories: *Intrinsic* and *Extrinsic* metrics. Intrinsic methods rely on inherent properties of the clustering results, while Extrinsic methods use external ground truths to infer the quality and effectiveness of clustering results. Such ground truth might be the labels of all data instances. Overall, evaluating the effectiveness of clustering methods and their results remains a complex task.

In a work focused on extrinsic clustering evaluation metrics, the authors introduced formal constraints on clustering evaluation metrics with the emphasis that such metrics should be intuitive, clarify limitations, formally provable, and discriminate metric families grouped by mathematical foundations (Amigó et al., 2009). The authors presented and motivated four constraints on quality measurements - homogeneity, completeness, rag bag, and size vs. quantity - and showed that only the extrinsic **B-CUBED** ($B^3$) metric (Bagga & Baldwin, 1998) out of many typically used metrics satisfies all four constraints; the others do not. These constraints are as follows: (1) *Homogeneity:* Clusters should not contain items from different categories. (2) *Completeness:* Items from the same category should be placed together in the same group. (3) *Rag Bag:* Disorder should be less detri-

mental in a disordered cluster than in a clean cluster. (4) *Cluster homogeneity:* A minor error in a large cluster should be preferred over a high number of small errors in small clusters.

Clustering involves two main components: the clustering method and the data representation. Different clustering algorithms identify different patterns and subgroups because they have different concepts of neighborhoods, assumptions about data distribution, strengths and weaknesses, and use different distance or similarity metrics. When the data representation is fixed, different clustering algorithms will produce different results, each with its degree of success. Common clustering algorithms include k-means (Ahmed et al., 2020), hierarchical clustering (Murtagh & Contreras, 2012), DBSCAN (Schubert et al., 2017), Gaussian mixture models (Reynolds et al., 2009), and spectral clustering (Von Luxburg, 2007). Alternatively, one can fix the clustering algorithm and train the data representation using techniques such as deep embedding with neural networks in deep clustering (Zhou et al., 2022; Caron et al., 2018; Bo et al., 2020). In this approach, a model learns how to optimally project and generate a data representation that is optimal for a given fixed clustering algorithm. In the case of inexact labels in a weakly supervised context, the clustering evaluation of the model should account for this uncertainty. Employing a sub-optimal cluster metric can result in over-optimizing the metric without considering the true structure of the data. In the case of coarse labels, this means without considering sub-labels. A poor metric can cause evaluation bias towards specific clusters instead of accurately representing the true structure of the data.

The $B^3$ algorithm is a precision and recall metric for clusters, as described by (Amigó et al., 2009). Two items that share a category are correctly related if and only if they occur in the same cluster. An item's $B^3$ precision is the fraction of objects in its cluster that share the item's category. The overall $B^3$ precision is calculated as the average precision of all items in the distribution. The $B^3$ recall is analogous. The $B^3$ algorithm can be used to numerically assess the quality of clustering assignments, but it does not account for imbalanced data sets. It is important to note that the $B^3$ algorithm presupposes that the ground truth labels are exact and that there are no (relevant) sub-clusters inside groups of equally labeled objects. For example, consider a data set with $n * m$ labels that are not visible beforehand, but instead, $m$ groups with $n$ item pairs are visible, and the aim is to identify the unknown number of sub-clusters. Two clustering algorithms are given: $C_A$, which finds $m$ super-sets, and $C_B$, which finds $n * m$ subsets. Although $C_B$ is preferred, the $B^3$ algorithm favors $C_A$ due to its higher completeness score. Consequently, the "completeness" attribute can be problematic in such circumstances. In other words, the constraint that "different clusters should contain items from different categories" (Amigó et al., 2009) can fail to select the correct model.

The key challenge in evaluating clustering quality is to construct a metric that provides a fair assessment for both balanced and imbalanced data sets while adjusting for label uncertainty. This is particularly challenging when the true structure of the data is unknown. Due to the lack of a ground-truth comparison, the process of breaking a set of clusters into multiple newer ones is fraught with "uncertainty." Assuming $m$ (possibly sub-) clusters and a deterministic aggregation function that unifies clusters into super-sets, by grouping clusters into (fewer) super-clusters, one can move from "uncertainty" closer to "certainty." This allows one to conclude: if the newly grouped clusters were of high quality, then the super-clusters are also "more likely" to be of high quality. Conversely, if the superclusters are "less likely" to have good quality, then so are the subclusters. Based on this motivation, a new method to evaluate clustering quality can be proposed, specifically addressing the term "completeness" in the conventional $B^3$ metric.

## 2 RELATED WORK

Weak supervision is a branch of machine learning in which the model is trained using noisy, incomplete, or inexact annotations instead of complete and accurate annotations (Zhou, 2018). A model is designed to deal with noise and uncertainty in annotations and make the best possible predictions based on the information given. Because producing high-quality labels is typically very costly, weak supervision is often used to generate additional cheaper, but lower-quality labeled data. Inaccurate training data contains defects or erroneous labels. The term "inexact" refers to training data with imprecise labels, such as coarse categories or probabilistic labels. Inexact supervision can result in a model with lesser prediction confidence, but also in misleading conclusions during supervised evaluation of clustering results. Cluster quality evaluation with inexact labels refers to the process of evaluating the performance of a clustering method when the ground truth labels are not

fully known or are too generic. There exist evaluation metrics that address cluster quality evaluation with inexact labels or uncertainty: Incomplete training data refers to data that lacks key information or characteristics. Data clustering with partial supervision, where data is neither completely nor accurately labeled, was also presented using a fuzzy clustering-based technique that uses available data knowledge to supervise the clustering process (Bouchachia & Pedrycz, 2006). The adjusted Rand Index (Rand, 1971) measures the similarity between the real and predicted (cluster) labels while adjusting for chance; related to accuracy. "Chance" refers to the possibility of achieving a specific outcome by random chance in the context. This involves considering the possibility that the agreement between the true and predicted labels might occur by coincidence, even if the clustering method is not appropriately grouping the data points. Normalized Mutual Information (Press et al., 2007) evaluates the mutual information between the real and predicted cluster labels, normalized by the entropy of both; where the entropy can be regarded as a measure of uncertainty. The Fowlkes-Mallows index (FMI) (Fowlkes & Mallows, 1983) computes the geometric mean of precision and recall between true and generated clusters. Uncertainty in the true labels might affect the accuracy of the clustering results because it may be difficult to accurately assign data points to their true labels if the actual labels are not clearly defined or known. FMI compensates for this uncertainty by calculating precision and recall using both the number of correct and incorrect predictions.

The $B^3$ clustering evaluation Amigó et al. (2009) is a metric that assesses each item's precision and recall in a data set. Precision is the ratio of items in the same cluster that belong to the same category as the item, while recall is the ratio of items in the item's category that are in its cluster. The final score is then often the harmonic mean of these individual scores. $B^3$ has gained a lot of attention and has received improvements and refinements to adjust to different situations. The adapted $B^3$ metrics (Moreno & Dias, 2015) were proposed for imbalanced data sets. The authors argue that the original family of $B^3$ metrics is not well adapted when data sets are imbalanced. The *Cluster-Identity-Checking Extended $B^3$* (CICE-$B^3$) (Rosales-Méndez & Ramírez-Cruz, 2013) was proposed as a new evaluation measure for overlapping clustering algorithms consisting of a new approach to determining precision, recall, and the F-measure, which analyzes object pairings that co-occur in clusters and/or classes. $B^3$ has also received criticism for overestimating performance because the clustering gets credit for putting an element in its own cluster van Heusden et al. (2022), which they repair by not counting the element itself.

This work emphasizes that the traditional $B^3$ metric may not provide accurate evaluation for clustering outcomes on finer subgroups or coarse labels. To address this limitation, a modified mathematical formula for $B^3$ is suggested. This modified formula incorporates a super aggregation of the cluster groups into its scoring function, aiming to improve the quality of the evaluation process.

## 3 BACKGROUND

Consider a data set $X$ consisting of elements $x_k$ with corresponding labels $y_k \in Y$. Let $C_j \subset X$ be a cluster, indexed by $j$, such that the data clusters are mutually disjoint, i.e. $\forall_{j,i} : C_i \cap C_j = \emptyset$. The set of all clusters be $C$. The union of all clusters is $\bigcup_{C_j \in C} C_j = X$, and so all elements belong to (exactly) one cluster. Let $\mathbf{1}_X : X \times X \to \{0, 1\}$ denote the indicator function for two elements $x_k, x_m \in X$. $\mathbf{1}_X$ returns a value of one iff both elements share the same label and belong to the same cluster; otherwise zero:

$$\mathbf{1}_X(x_k, x_m) := \begin{cases} 1 & \text{if } y_k = y_m \wedge \exists_{C_j \in C} : x_k, x_m \in C_j \ , \\ 0 & \text{otherwise.} \end{cases} \tag{1}$$

The $B^3$ cluster score - based on precision, recall, and the $F_\beta$-score - then is (Amigó et al., 2009):

$$\begin{aligned} P(X) &:= \mathbb{E}_{x_k}\left[\mathbb{E}_{x_m | \exists_{C_j \in C} : x_k, x_m \in C_j}[\mathbf{1}_X(x_k, x_m)]\right] \\ R(X) &:= \mathbb{E}_{x_k}\left[\mathbb{E}_{x_m | y_k = y_m}[\mathbf{1}_X(x_k, x_m)]\right] \\ B^3(X) &:= F_\beta(X) \triangleq \frac{(1 + \beta^2)P(X)R(X)}{\beta^2 P(X) + R(X)} \end{aligned} \tag{2}$$

## 4 METHODS

When attempting to identify sub-clusters within coarse ground truth categories, the assumption that elements belonging to the same super-category should be grouped ("completeness") can be problematic. We propose a modified evaluation metric called $\alpha$Max-$B^3$. The core concept involves combining clusters based on their most frequently occurring label and consolidating them into label-aggregated new cluster sets. $\alpha$Max-$B^3$ is calculated over a score weighting over the consolidated sets and the original clusters, utilizing a trade-off parameter $\alpha$ for ground-truth uncertainty.

First, for a $C_j$, we define the most frequent label of its elements as the cluster label:

$$\hat{y}_j := \operatorname*{argmax}_{y \in Y} |\{x_k \in C_j : y_k = y\}| \tag{3}$$

Next, super-sets $S_y$ are generated by merging clusters with equal max-pooled cluster labels $\hat{y}$:

$$S_y := \bigcup_{C_j \in C} \{x_k \in C_j : \hat{y}_j = y\} \tag{4}$$

Thus, given $|Y|$ labels, at most $|Y|$-many new super-sets $S_y$ are generated. A corresponding super-set for a particular label may be empty if it never reaches the majority within any cluster, in which case it is just disregarded and ignored. In the unlikely event that argmax is not unique, the cluster is simply *not* merged, left as it is, and doesn't count as a new super-set. The final $\alpha$Max-$B^3$ scores are then (a weighted version of) the $B^3$ scores between the new (non-empty) super-sets $\hat{S} := \{S_y : S_y \neq \emptyset\}$, and the original cluster sets $C$. Let $P_{[y]}(C_j)$ denote the $B^3$ precision score of label $y$ on elements in a cluster $C$, and $R_{[y]}(C_j)$ respectively that of the recall. Let $K$ be a set of cluster indices. We denote the weighted average precision score over multiple clusters as $\mathbb{E}[\bigwedge_{k \in K} P_{[y]}(C_k)] := \sum_{k \in K} |C_k|_y P_{[y]}(C_k) / |\bigcup_{k \in K} C_k|_y$; the weighted recall score is defined analogously.

**Theorem 1.** *Let $C_i, C_j$ be disjoint clusters. Then:* $\mathbb{E}[P_{[y]}(C_i) \wedge P_{[y]}(C_j)] \geq P_{[y]}(C_i \cup C_j)$.

Theorem 1 (Proof A.1) shows that the precision score of the super-set is always less than or equal to the average precision score of the individual subsets. This implies a *monotonically* decreasing relationship in precision, which is crucial as it yields no fluctuations in score translations. A "decrease" in precision scores is not concerning because the actual cluster assignments of the elements remain unchanged; only additional super-clusters are created. The evaluation of the cluster assignments is done on a different "scale"; whether scores are higher or lower in comparison is not directly relevant, but having a smooth monotonic transition is important. A stricter definition of Theorem 1 is:

**Proposition 1.** $\mathbb{E}[P_{[y]}(C_i) \wedge P_{[y]}(C_j)] = P_{[y]}(C_i \cup C_j) \iff P_{[y]}(C_i) = P_{[y]}(C_j)$.

This means that the average of the precision scores of two clusters for a given label $y$ will only be equal to that of their super-cluster iff the precisions of both sub-clusters are the same (Proof A.2). It can be directly inferred from Theorem 1 and Proposition 1 that the translation is *strictly* monotonically decreasing when $P_{[y]}(C_i) \neq P_{[y]}(C_j)$, as we established both a "less equal" and an "equality" relationship. Without equality, Theorem 1 reduces to "strictly less" (Proof A.3).

**Corollary 1.** $\mathbb{E}[P_{[y]}(C_i) \wedge P_{[y]}(C_j)] > P_{[y]}(C_i \cup C_j) \iff P_{[y]}(C_i) \neq P_{[y]}(C_j)$.

Analogously, let $R_{[y]}(C)$ denote recall scores of label $y$ on elements in a cluster $C_j$.

**Theorem 2.** *Let $C_i, C_j$ be two disjoint clusters. Then:* $\mathbb{E}[R_{[y]}(C_i) \wedge R_{[y]}(C_j)] \leq R_{[y]}(C_i \cup C_j)$.

This, again, yields a monotonic score translation, but this time, the $B^3$ recall is higher on the super-set (Proof A.4). A stricter definition on Theorem 2 yields:

**Proposition 2.** $\mathbb{E}[R_{[y]}(C_i) \wedge R_{[y]}(C_j)] = R_{[y]}(C_i \cup C_j) \iff R_{[y]}(C_i) = 0 \vee R_{[y]}(C_j) = 0$.

And similarly, Theorem 2 and Proposition 2 imply the translation is *strictly* monotonically increasing whenever $R_{[y]}(C_i) \neq 0 \wedge R_{[y]}(C_j) \neq 0$ (Proof A.5).

**Corollary 2.** $\mathbb{E}[R_{[y]}(C_i) \wedge R_{[y]}(C_j)] < R_{[y]}(C_i \cup C_j) \iff R_{[y]}(C_i) \neq 0 \wedge R_{[y]}(C_j) \neq 0$.

Thus, the translation of the super-set (precision and recall) scores is strictly monotonic - respectively decreasing and increasing. However, the $F_\beta$ measure, which is a composite function involving sums and products of both decreasing and increasing monotonic functions, does not necessarily exhibit monotonicity. While Theorems 1 and 2 refer to pairwise clusters, this also holds for any number of clusters, as can be proven by merging pairwise clusters inductively (Proof A.6).

Let $\mathbb{S}$ be the collection of all clusters. A super-set $S_i \subset \mathbb{S}$ is the union of a specific list of clusters indexed by $C_j$, i.e., $S_i := \bigcup_{j \in \mathcal{J}} C_j$, where $\mathcal{J}$ represents the set of cluster indices. Each $C_j$ is always assigned to exactly one $S_i$. The set of all identified super-clusters is denoted by $\hat{S}$. Elements in $\hat{S}$ are mutually disjoint. The number of elements in a collection is denoted by $|S_i|, |C_j|$, and $|S_i|_y, |C_j|_y$ is the number of elements with label $y$ in the respective collection; $|y|$ be the total number of all elements with label $y$. Naively, one may now want to assign weights to the clusters based on their cardinality. Since each cluster index $j$ is uniquely associated with a super-set index $i$, using $j$ is sufficient to represent the weight. Let $\eta^{[j]}$ be the weight assigned to cluster $C_j$, which is calculated as the ratio of the number of elements in $C_j$ to the total number of elements in $S_i$. Using this naive cluster weight, we can derive the following results.

$$\eta^{[j]} := \frac{|C_j|}{|S_i|} = \frac{|C_j|}{|S_i|} \cdot \frac{|y||S_i|_y|C_j|_y}{|y||S_i|_y|C_j|_y} = \frac{\frac{|S_i|_y}{|S_i|}}{\frac{|C_j|_y}{|C_j|}} / \frac{\frac{|S_i|_y}{|y|}}{\frac{|C_j|_y}{|y|}} = \frac{P_{[y]}(S_i)}{P_{[y]}(C_j)} / \frac{R_{[y]}(S_i)}{R_{[y]}(C_j)} \tag{5}$$

Hence, we can express the ratio $\eta^{[j]}$ of the cluster size to its super-set in terms of precision and recall. Theorems 1, 2 imply $\mathbb{E}_{C_j \in \mathbb{S}_i}[\frac{P_{[y]}(C_j)}{P_{[y]}(S_i)}] \geq 1 \iff P_{[y]}(S_i) \leq \mathbb{E}[P_{[y]}(C_j)]$; and $\frac{R_{[y]}(C_j)}{R_{[y]}(S_i)} \leq 1 \iff R_{[y]}(C_j) \leq R_{[y]}(S_i)$, given $|C_j|_y \leq |S_i|_y$ (Proof A.9); implying $\mathbb{E}[\eta^{[j]}] \leq 1$ and the equivalence:

$$\mathbb{E}[\eta^{[j]}] = \mathbb{E}\left[\frac{P_{[y]}(S_i)R_{[y]}(C_j)}{P_{[y]}(C_j)R_{[y]}(S_i)}\right] = \mathbb{E}\left[\frac{P_{[y]}(S_i)}{P_{[y]}(C_j)} \frac{\min[R_{[y]}(S_i), R_{[y]}(C_j)]}{\max[R_{[y]}(S_i), R_{[y]}(C_j)]}\right]$$
$$= \mathbb{E}\left[\frac{\min[P_{[y]}(S_i), P_{[y]}(C_j)]}{\max[P_{[y]}(S_i), P_{[y]}(C_j)]}\right] \cdot \frac{\min[R_{[y]}(S_i), R_{[y]}(C_j)]}{\max[R_{[y]}(S_i), R_{[y]}(C_j)]} \leq 1 \tag{6}$$

Note that $\mathbb{E}[\eta^{[j]}] \leq 1 \not\Longrightarrow \forall_j : \eta^{[j]} \leq 1$; Let $\alpha \in [0, 1]$ be a cluster-specific local uncertainty indicator. We can write a slightly modified non-monotonic $\alpha$-weighted expression of Equation 6 as:

$$\eta_\alpha^{[j]} = \frac{\min[P_{[y]}(S_i), \alpha P_{[y]}(C_j)]}{\max[P_{[y]}(S_i), \alpha P_{[y]}(C_j)]} \cdot \frac{\min[\alpha R_{[y]}(S_i), R_{[y]}(C_j)]}{\max[\alpha R_{[y]}(S_i), R_{[y]}(C_j)]} \tag{7}$$

where it holds $0 \leq \eta_\alpha^{[j]} \leq 1$, with $\lim_{\alpha \to 0} \mathbb{E}[\eta_\alpha^{[j]}] = 0^+$ and $\lim_{\alpha \to 1} \mathbb{E}[\eta_\alpha^{[j]}] = \mathbb{E}[\eta^{[j]}]^+$; both approaching from above. Therefore, it can be either $\eta_\alpha^{[j]} \leq \mathbb{E}[\eta^{[j]}]$ or $\eta_\alpha^{[j]} \geq \mathbb{E}[\eta^{[j]}]$ for different $\alpha$. Thus, non-monotonic (see Proof A.8). In fact, it is $\min(p, \alpha q) = \max(p, \alpha q) \iff \alpha = p/q$.

For arbitrary but *fixed* precision and recall scores, the function $\eta_\alpha^{[j]}$ over $\alpha$ has two "inversion points": $p_1 := \frac{P_{[y]}(S_i)}{P_{[y]}(C_j)}$ and $p_2 := \frac{R_{[y]}(C_j)}{R_{[y]}(S_i)}$; so $\forall_{\min(p_1,p_2) \leq \alpha \leq \alpha' \leq \max(p_1,p_2)} : \eta_\alpha^{[j]} = \eta_{\alpha'}^{[j]}$, which means that given fixed scores of precision and recall, $\eta_\alpha^{[j]}$ has a function maxima plateau of score uncertainty in the interval $[\min(p_1, p_2), \max(p_1, p_2)]$ (Proof A.10).

The weight function $\eta_\alpha$ over $\alpha$ for sample values is plotted in Figure 1. It reaches a maximum at the value of one if: $\max \eta_\alpha^{[j]} = 1 \iff \frac{P_{[y]}(S_i)}{P_{[y]}(C_j)} = \frac{R_{[y]}(C_j)}{R_{[y]}(S_i)}$. The local uncertainty score $\eta_\alpha^{[j]}$ can be seen as the "super-set cluster importance" which weighs how important the score of the corresponding super-set is over the original cluster $C_j$. E.g., when

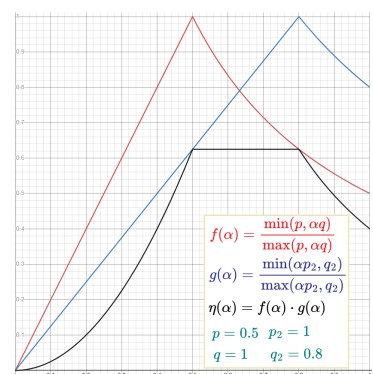

Figure 1: Graph of $\eta_\alpha^{[j]}$ over $\alpha$.

the local uncertainty is zero, then $\alpha$-Max-$B^3$ is equivalent to $B^3$. Although $\alpha$ can be set manually, we assume no prior knowledge on label uncertainty and set $\alpha := \min(p_1, p_2)$ as the default choice for extracting the highest possible $\eta_\alpha^{[j]}$ weight with minimal uncertainty. Let $Q \in \{C, S\}, x_k \in C_j$. We write $Q_{k|q} \equiv x_k \in Q_q$ and $k|j \equiv j$. The final $\alpha$Max-$B^3$ score is the $F_\beta$ score using:

$$
\begin{aligned}
P_\alpha(X) &\triangleq \frac{1}{|X|} \sum_{x_k \in X} \eta_\alpha^{[k|j]} P(S_{k|i}) + (1 - \eta_\alpha^{[k|j]}) P(C_{k|j}) \\
R_\alpha(X) &\triangleq \frac{1}{|X|} \sum_{x_k \in X} \eta_\alpha^{[k|j]} R(S_{k|i}) + (1 - \eta_\alpha^{[k|j]}) R(C_{k|j})
\end{aligned}
\tag{8}
$$

### 4.1 EXTENSION TO IMBALANCED DATA SETS

Imbalanced class distribution is problematic during evaluation, as a simple averaging of $F_\beta$ scores disregards label imbalance. Conventional weighted averaging, which assigns weights inverse to frequency, may also not be appropriate because it fails to account for the diminishing value of newly added data points as the number of items increases. This is due to the overlap in data information and the fact that new data points are more likely to be close copies of existing ones when the volume of samples is large; known as the *effective number of samples* (ENS) (Cui et al., 2019), which measures the volume of a collection of $n$ samples given a hyper-parameter $\delta \in [0, 1)$ using the derived formula $v_\delta(n) := (1 - \delta^n)/(1 - \delta)$ for inverse weighting. Let $|y_k|$ denote the frequency of label a $y_k \in Y$ in the data set. The normalized inverse weight is $w_\delta(x_k) := v_\delta(|y_k|)^{-1} / \sum_{y_i}^Y v_\delta(|y_i|)^{-1}$; and $\delta$ is as a hyper-parameter. (Cui et al., 2019) proposed a unified default value of $\delta = (|X| - 1)/|X|$; which we will be adopting. $w_\delta(|y|)$ is then incorporated to define $\alpha$Max-$B_\delta^3$:

$$
\begin{aligned}
P_\alpha^{[\delta]}(X) &\triangleq \sum_{x_k \in X} \frac{1}{|y_k|} w_\delta(x_k) P_\alpha(x_k) \\
R_\alpha^{[\delta]}(X) &\triangleq \sum_{x_k \in X} \frac{1}{|y_k|} w_\delta(x_k) R_\alpha(x_k)
\end{aligned}
\tag{9}
$$

where $P_\alpha(x_k) := \eta_\alpha^{[k|j]} P(S_{k|i}) + (1 - \eta_\alpha^{[k|j]}) P(C_{k|j})$ is the precision score of an *element*; $R_\alpha(x_k)$ is defined analogously for recall. As mentioned in (Cui et al., 2019), $\delta = 0$ corresponds to no re-weighting, thus, $\delta = 0 \implies P_\alpha^{[\delta]}(X) = P_\alpha(X) \wedge R_\alpha^{[\delta]}(X) = R_\alpha(X)$; while $\delta \to 1$ approaches re-weighting on inverse class frequency. Theorems 1 and 2 remain true also for the imbalanced $\delta$-weighted version of precision $P_{[y]}^\delta$ and recall $R_{[y]}^\delta$. The case $\delta = 0$ is trivial, but it also holds for any fixed $\delta \in [0, 1)$, i.e. $P_{[y]}^\delta(C_i) + P_{[y]}^\delta(C_j) \geq P_{[y]}^\delta(C_i \cup C_j)$ and $R_{[y]}^\delta(C_i) + R_{[y]}^\delta(C_j) \leq R_{[y]}^\delta(C_i \cup C_j)$. This becomes evident upon canceling out the weighting factor $\delta$ on respectively both sides of the inequality (Proof A.7). Though essentially an extension of the other, the $\delta$-weighted version has been independently addressed for conceptual clarity. Since the $\eta_\alpha^{[k|j]}$ terms get canceled, this implies that any other weighting function can be used in place of the ENS and the theorems still hold.

## 5 EXPERIMENTAL EVALUATION

### 5.1 COMPARISON OF $\alpha$MAX-B3 AND TRADITIONAL B3 ON SYNTHETIC DATA

The proposed $\alpha$Max-$B^3$ method was tested on a synthetic data set with five separable classes and compared to the traditional $B^3$ using $KMeans$ for multiple values of $k$ clusters. Both methods performed best for $k = 5$ and equally for $k \leq 5$. However, for $k > 5$, the proposed method provided a more robust and fair evaluation, assigning higher scores to sub-clusters and reasonable sub-groups than the traditional $B^3$.

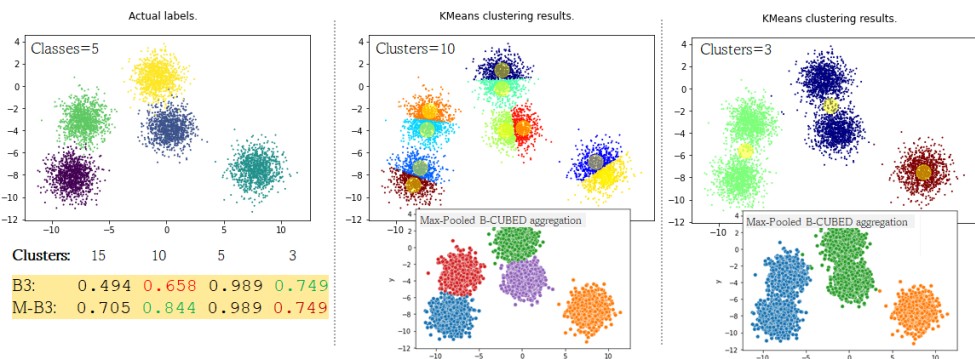

Figure 2: Multiple clustering assignments are evaluated using clustering scores and visual illustrations. The optimal clustering consists of five classes, where a finer and purer sub-clustering should have better scores compared than a coarser and less pure clustering. The $\alpha$Max-$B^3$ metric, a variation of the standard $B^3$, generates more robust and fair scores. It prioritizes correct extraction of sub-clusters over incorrect super-clusters, while also considering non-homogeneous super-clusters only when the number of sub-clusters becomes excessive.

## 5.2 IMBALANCED DATA SET

We repeated the above experiment with imbalanced class labels (ratio 1:2:4:8:16) instead of balanced, to assess the metric in an imbalanced data set scenario to analyze $\alpha$Max-$B_\delta^3$ (Figure 3).

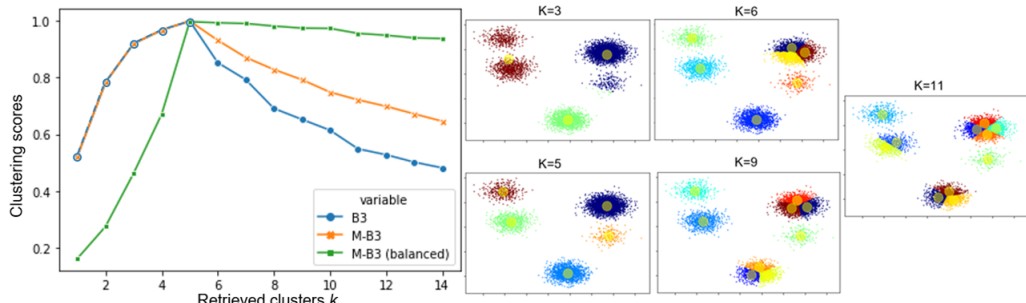

Figure 3: The $B^3$, $\alpha$Max-$B^3$, and $\alpha$Max-$B_\delta^3$ scores on an imbalanced data set with five labels. The performance was evaluated based on different cluster counts $k$. On coarse clusters ($k \leq 5$), $B^3$ and $\alpha$Max-$B^3$ perform equally. However, $\alpha$Max-$B^3$ performs fairer on finer sub-clusters ($k \geq 5$) since it gives more weight to sub-clusters under uncertainty. $\alpha$Max-$B_\delta^3$ accounts for class imbalance; which results in differences in $k < 5$.

$B^3$ and $\alpha$Max-$B^3$ have identical scores for $k \leq 5$ as expected, but the $\delta$-balanced variant returned a much lower score, preferring sub-clusters in the most frequent labels and keeping clusters with few points together. This is consistent with that there is more uncertainty in splitting smaller clusters with few points than larger clusters with many points, because the more points there are, the more likely it is that further sub-groups exist. $\delta$ was chosen based on the default recommendation (Cui et al., 2019); but other values could have also been used to tweak the balancing.

## 5.3 EVALUATION OF AUTOMATIC UNCERTAINTY DETERMINATION

We analyzed the automatic uncertainty determination of $\alpha$ in our method. To empirically demonstrate that the automatically determined values of the uncertainty parameter $\alpha$ relate to the actual ground truth uncertainty, we compared our method to the standard B3 evaluation score and the benchmark scores of the real uncertainty value $\alpha$. We used artificial clustering problems with pure and noise groups of cluster assignments, both with and without miss-assignments or outliers. Hence, the actual number of subclusters, which is equivalent to the uncertainty, provided a ground truth benchmark. We set $k = 10$ classes and evaluated the $B^3$ score for $n \in \{1, ..., 8\}$ subclusters per class, requiring a data set of $k \times 8!$ instances and a total of $k \times n$ clusters. The ground truth benchmark was thus using the uncertainty of $\alpha = 1/n$. The results are given in Table 1. As can be seen,

the scores for the automatically determined values of $\alpha$ match those of the ground truth uncertainty perfectly without any noise, and almost perfectly up to about four decimals with increasing noise. In particular, different from those of the traditional $B^3$.

Table 1: A comparison of scores against ground truth uncertainty.

| | NO NOISE / PURE CLUSTERS | | | 25% RANDOM LABELS | | | 50% RANDOM LABELS | |
|---|---|---|---|---|---|---|---|---|
| N | B3 | $\alpha$ B3 | $\frac{1}{n}$ B3 | N | B3 | $\alpha$ B3 | $\frac{1}{n}$ B3 | N | B3 | $\alpha$ B3 | $\frac{1}{n}$ B3 |
| 1 | 1.00 | 1.00000 | 1.00000 | 1 | 0.61 | 0.60591 | 0.60591 | 1 | 0.32 | 0.32439 | 0.32439 |
| 2 | 0.67 | 0.85714 | 0.85714 | 2 | 0.41 | 0.52126 | 0.52126 | 2 | 0.22 | 0.27938 | 0.27938 |
| 3 | 0.50 | 0.71429 | 0.71429 | 3 | 0.30 | 0.43281 | 0.43280 | 3 | 0.16 | 0.23241 | 0.23240 |
| 4 | 0.40 | 0.60870 | 0.60870 | 4 | 0.24 | 0.36957 | 0.36955 | 4 | 0.13 | 0.19755 | 0.19754 |
| 5 | 0.33 | 0.52941 | 0.52941 | 5 | 0.20 | 0.32099 | 0.32098 | 5 | 0.11 | 0.17234 | 0.17231 |
| 6 | 0.29 | 0.46809 | 0.46809 | 6 | 0.17 | 0.28405 | 0.28403 | 6 | 0.09 | 0.15187 | 0.15185 |
| 7 | 0.25 | 0.41935 | 0.41935 | 7 | 0.15 | 0.25339 | 0.25338 | 7 | 0.08 | 0.13623 | 0.13620 |
| 8 | 0.22 | 0.37975 | 0.37975 | 8 | 0.13 | 0.23050 | 0.23048 | 8 | 0.07 | 0.12344 | 0.12340 |

We further performed experiments on data sets exhibiting mixed subgroup uncertainty, defined as the union of two different data sets of equal cardinality, each characterized by distinct subgroup uncertainty. The benchmark was established as the arithmetic mean of both subgroups' uncertainties; we denote $\varnothing\frac{1}{n}$ B3. The results are presented in Table 2. When merging data sets of equal size but with varying numbers of subclusters, the scores of the automatic evaluation remained similar, albeit with some notable differences. Increasing the number of samples and the number of class clusters $k$ only reduced the score difference slightly.

Table 2: A comparison of scores from merged data with varying levels of uncertainty (subcluster tuples) under different degrees of noise, and number of instances and cluster classes.

| | NO NOISE / PURE CLUSTERS | | | 25% RANDOM LABELS | | | 50% RANDOM LABELS | |
|---|---|---|---|---|---|---|---|---|
| N | B3 | $\alpha$ B3 | $\varnothing\frac{1}{n}$ B3 | N | B3 | $\alpha$ B3 | $\varnothing\frac{1}{n}$ B3 | N | B3 | $\alpha$ B3 | $\varnothing\frac{1}{n}$ B3 |
| (1,5) | 0.559 | 0.673 | 0.672 | (1,5) | 0.340 | 0.411 | 0.410 | (1,5) | 0.186 | 0.226 | 0.225 |
| (2,6) | 0.356 | 0.512 | 0.481 | (2,6) | 0.217 | 0.312 | 0.295 | (2,6) | 0.120 | 0.173 | 0.164 |
| (3,7) | 0.269 | 0.416 | 0.384 | (3,7) | 0.165 | 0.255 | 0.236 | (3,7) | 0.091 | 0.141 | 0.133 |
| (4,8) | 0.219 | 0.353 | 0.323 | (4,8) | 0.135 | 0.217 | 0.200 | (4,8) | 0.075 | 0.121 | 0.113 |

| | K=50 | | | K=150 | | | K=300 | |
|---|---|---|---|---|---|---|---|---|
| N | B3 | $\alpha$ B3 | $\varnothing\frac{1}{n}$ B3 | N | B3 | $\alpha$ B3 | $\varnothing\frac{1}{n}$ B3 | N | B3 | $\alpha$ B3 | $\varnothing\frac{1}{n}$ B3 |
| (1,5) | 0.467 | 0.622 | 0.620 | (1,5) | 0.450 | 0.609 | 0.608 | (1,5) | 0.445 | 0.606 | 0.604 |
| (2,6) | 0.290 | 0.440 | 0.422 | (2,6) | 0.279 | 0.427 | 0.410 | (2,6) | 0.276 | 0.423 | 0.407 |
| (3,7) | 0.217 | 0.350 | 0.327 | (3,7) | 0.208 | 0.338 | 0.317 | (3,7) | 0.206 | 0.335 | 0.314 |
| (4,8) | 0.176 | 0.294 | 0.273 | (4,8) | 0.168 | 0.283 | 0.264 | (4,8) | 0.166 | 0.281 | 0.261 |

## 5.4 UNCERTAINTY ESTIMATE AND EXTRAPOLATION

We extract the level of uncertainty by measuring the plateau interval and extrapolating the $\alpha$ values and compare the estimation with the ground truth uncertainty of the data set. We do this, by calculating the expectation of all automatically determined alpha values (over all function Plateaus intervals using $\alpha := \min(p_1, p_2)$; see Figure 1). The results are given in Table 3; for a single and merged data sets respectively (as in Section 5.3):

On the consistent data set, the uncertainty estimation was perfect. However, on merged data sets with inconsistent (i.e. different) uncertainties, there were slight differences, but the estimation was nevertheless close.

Table 3: Estimating the data set uncertainty by extrapolating $\alpha$.

| CONSISTENT DATA SET | | | | | | | | | MERGED DATA SETS | | | |
|---|---|---|---|---|---|---|---|---|---|---|---|---|
| REAL $\alpha$ | 1 | $\frac{1}{2}$ | $\frac{1}{3}$ | $\frac{1}{4}$ | $\frac{1}{5}$ | $\frac{1}{6}$ | $\frac{1}{7}$ | $\frac{1}{8}$ | REAL $\alpha$ | 0.6 | 0.333 | 0.238 | 0.188 |
| EXTR. $\alpha$ | 1 | $\frac{1}{2}$ | $\frac{1}{3}$ | $\frac{1}{4}$ | $\frac{1}{5}$ | $\frac{1}{6}$ | $\frac{1}{7}$ | $\frac{1}{8}$ | EXTR. $\alpha$ | 0.5 | 0.290 | 0.218 | 0.172 |

## 6 DISCUSSION

Coarse ground-truth labels make it harder to accurately evaluate clustering results. A poor assessment measure can certainly confuse the analyst and lead to incorrect interpretations and conclusions (e.g. the choice of the right model, hyper-parameters, etc.). Being aware of the issue enables us to counteract or avert this situation. The fundamental challenge is how to adequately evaluate clustering results using a supervised metric if the labels are coarse and not representative as ground-truths. We approached this problem by leveraging the observation that we can determine or choose a degree of uncertainty to either encourage or discourage sub-group identification. Even though the labels are inexact, using a supervised measure is still required in this scenario. An unsupervised loss function is often based on the assumption that data points within a cluster are similar to each other but dissimilar to those in other clusters. This may not be appropriate since sub-groups could still be extremely similar, but more importantly, it does not guarantee that the clusters are homogeneous, especially as we do not want to mix instances with different labels in the same cluster. If the labels were exactly the ground truth, or if we were not concerned with discovering sub-groups, setting the uncertainty $1 - \alpha$ to zero will yield the same results as $B^3$.

The $\alpha$ parameter in $\alpha$Max-$B^3$ is cluster-specific, unlike $\delta$, because it is dependent on the precision and recall of a particular cluster. $\alpha$ and $\eta_\alpha$ are related, although they have distinct roles. The final cluster uncertainty score is based on precision and recall and reflected by $\eta_\alpha$. On the other hand, $\alpha$ reflects the uncertainty between the original clusters and the merged super-clusters. $\eta_\alpha$ then uses $\alpha$ for its score assessment. The values for $\alpha$ and $\delta$ can also be set manually. However, manual adjustment introduces hyper-parameters, which can be time-consuming and tricky to optimize.

The automatic determination of $\alpha$, selected over the function's maxima Plateau (see Figure 1), approximates the real (unknown) uncertainty in the data very well, and about perfectly over a consistent set of uncertainty across all clusters. However, the more noisy and uneven the cluster uncertainty is across the clusters (i.e. different levels of uncertainties across clusters), the less accurate this approximation becomes. Overall, this is very fortunate and provides a good automatic way of setting the uncertainty. Furthermore, we can also extract the level of uncertainty by measuring the plateau interval and extrapolating the $\alpha$ values to obtain an approximation or estimate of the data set uncertainty.

## 7 SUMMARY & CONCLUSION

This work has identified a concern regarding the widely used $B^3$ cluster quality metric when coarse labels are involved, which can lead to unfair and misleading evaluations. To address this issue, a new metric called $\alpha$Max-$B^3$ has been proposed. This solution modifies the evaluation of the standard $B^3$ method by adapting to sub-group uncertainty in ground-truth labels and can be generalized to accommodate imbalanced data sets. The proposed evaluation method merges clusters into larger groups called super-sets and evaluates them using a modified $B^3$ based metric that applies a weighting factor to control the contribution of the super-sets. Unlike the standard $B^3$ technique, $\alpha$Max-$B^3$ can produce more robust and fair results and adapt to label uncertainty. This uncertainty is controlled by an $\alpha$ parameter and setting it to zero yields the same results as the standard $B^3$ metric. Our solution is easy to implement, has a solid theoretical foundation, and has many practical applications, making it an attractive evaluation metric in the field of clustering.

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

# A PROOFS

## A.1 THEOREM 1

Let $C_i, C_j$ be disjoint clusters of distinct elements. Theorem 1 is to be proven: $\mathbb{E}[P_{[y]}(C_i) \wedge P_{[y]}(C_j)] \geq P_{[y]}(C_i \cup C_j)$, where $P_{[y]}(C_j)$ denotes the precision scores of label $y$ on elements in a cluster $C_j$. Let $|C_i|, |C_j| \in \mathbb{N}$ denote the total number of elements in $C_i, C_j$ and $|C_i|_y, |C_j|_y \in \mathbb{N}$ respectively the number of elements in $C_i, C_j$ with label $y$. We know $0 \leq |C_j|_y \leq |C_j| \wedge 0 \leq |C_i|_y \leq |C_i|$. Therefore:

$$
\begin{aligned}
& \mathbb{E}[P_{[y]}(C_i) \wedge P_{[y]}(C_j)] \geq P_{[y]}(C_i \cup C_j) \\
\iff\;& \frac{|C_i|_y P_y(C_i) + |C_j|_y P_y(C_j)}{|C_i|_y + |C_j|_y} \geq P_y(C_i \cup C_j) \\
\iff\;& \frac{|C_i|_y \frac{|C_i|_y}{|C_i|} + |C_j|_y \frac{|C_j|_y}{|C_j|}}{|C_i|_y + |C_j|_y} \geq \frac{|C_i|_y + |C_j|_y}{|C_i| + |C_j|} \\
\iff\;& \frac{\frac{|C_i|_y^2}{|C_i|} + \frac{|C_j|_y^2}{|C_j|}}{|C_i|_y + |C_j|_y} - \frac{|C_i|_y + |C_j|_y}{|C_i| + |C_j|} \geq 0 \\
\iff\;& \frac{|C_j||C_i|_y^2 + |C_i||C_j|_y^2}{|C_i||C_j|(|C_i|_y + |C_j|_y)} - \frac{|C_i|_y + |C_j|_y}{|C_i| + |C_j|} \geq 0 \\
\iff\;& \frac{(|C_i||C_j|_y - |C_j||C_i|_y)^2}{|C_i||C_j|(|C_i|_y + |C_j|_y)(|C_i| + |C_j|)} \geq 0 \\
\iff\;& (|C_i||C_j|_y - |C_j||C_i|_y)^2 \geq 0 \iff \top
\end{aligned}
\tag{10}
$$

## A.2 PROPOSITION 1

We prove statement Proposition 1: $\mathbb{E}[P_{[y]}(C_i) \wedge P_{[y]}(C_j)] = P_{[y]}(C_i \cup C_j) \iff P_{[y]}(C_i) = P_{[y]}(C_j)$, where $P_{[y]}(C_j)$ is the precision scores of label $y$ on elements in a cluster $C_j$. Let $0 \leq |C_j|_y \leq |C_j| \wedge 0 \leq |C_i|_y \leq |C_i|$ be defined as in Proof A.1, from which we know the inequality: $\mathbb{E}[P_{[y]}(C_i) \wedge P_{[y]}(C_j)] \geq P_y(C_i \cup C_j) \iff (|C_i|_y|C_j| - |C_j|_y|C_i|)^2 \geq 0$, and so:

$$
\begin{aligned}
& \mathbb{E}[P_{[y]}(C_i) \wedge P_{[y]}(C_j)] = P_{[y}(C_i \cup C_j) \\
\iff\;& (|C_i|_y|C_j| - |C_j|_y|C_i|)^2 = 0 \\
\iff\;& \frac{|C_i|_y}{|C_i|} = \frac{|C_i|_y}{|C_i|} \iff P_{[y]}(C_i) = P_{[y]}(C_j)
\end{aligned}
\tag{11}
$$

## A.3 COROLLARY 1

We prove Corollary 1, which states that $\mathbb{E}[P_{[y]}(C_i) \wedge P_{[y]}(C_j)] > P_{[y]}(C_i \cup C_j) \iff P_{[y]}(C_i) \neq P_{[y]}(C_j)$. According to Theorem 1, $\mathbb{E}[P_{[y]}(C_i) \wedge P_{[y]}(C_j)] \geq P_{[y]}(C_i \cup C_j)$ always holds. Therefore,

if $P_{[y]}(C_i) \neq P_{[y]}(C_j)$, we apply Proposition 1 to conclude $\mathbb{E}[P_{[y]}(C_i) \wedge P_{[y]}(C_j)] \neq P_{[y]}(C_i \cup C_j)$. Consequently, it must follow that $\mathbb{E}[P_{[y]}(C_i) \wedge P_{[y]}(C_j)] > P_{[y]}(C_i \cup C_j)$.

## A.4 THEOREM 2

Let $C_i, C_j$ be disjoint clusters of distinct elements. Theorem 2 is to be proven: $\mathbb{E}[R_{[y]}(C_i) \wedge R_{[y]}(C_j)] \leq R_{[y]}(C_i \cup C_j)$, where $R_{[y]}(C_j)$ denotes the $B^3$ recall score of label $y$ on elements in a cluster $C_j$. Let $|y| \in \mathbb{N}$ denote the total number of elements with label $y$ in all clusters, i.e. in the entire data set, and $|C_i|_y, |C_j|_y \in \mathbb{N}$ respectively the number of elements in clusters $C_i, C_j$ with label $y$. Then:

$$
\begin{aligned}
& \mathbb{E}[R_{[y]}(C_i) \wedge R_{[y]}(C_j)] \leq R_{[y]}(C_i \cup C_j) \\
\iff & \frac{|C_i|_y R_{[y]}(C_i) + |C_j|_y R_{[y]}(C_j)}{|C_i|_y + |C_j|_y} \leq R_{[y]}(C_i \cup C_j) \\
\iff & \frac{(|C_i|_y \frac{|C_i|_y}{|y|}) + (|C_j|_y \frac{|C_j|_y}{|y|})}{|C_i|_y + |C_j|_y} \leq \frac{|C_i|_y + |C_j|_y}{|y|} \\
\iff & 0 \leq \frac{|C_i|_y + |C_j|_y}{|y|} - \frac{|C_i|_y^2 + |C_j|_y^2}{|y|(|C_i|_y + |C_j|_y)} \\
\iff & 0 \leq \frac{(|C_i|_y + |C_j|_y)^2 - |C_i|_y^2 - |C_j|_y^2}{|y|(|C_i|_y + |C_j|_y)} \\
\iff & 0 \leq \frac{2|C_i|_y |C_j|_y}{|y|(|C_i|_y + |C_j|_y)} \\
\iff & 0 \leq |C_i|_y |C_j|_y \iff \top
\end{aligned}
\tag{12}
$$

## A.5 PROPOSITION 2

We prove Proposition 2: $\mathbb{E}[R_{[y]}(C_i) \wedge R_{[y]}(C_j)] = R_{[y]}(C_i \cup C_j) \iff R_{[y]}(C_i) = 0 \vee R_{[y]}(C_j) = 0$. $R_{[y]}(C_j)$ is the $B^3$ recall score of label $y$ on elements in a cluster. Let $|y|, C_i, C_j, |C_i|_y, |C_j|_y$ be as in Proof A.1. We know $\mathbb{E}[R_{[y]}(C_i) \wedge R_{[y]}(C_j)] \leq R_{[y]}(C_i \cup C_j) \iff |C_i|_y |C_j|_y \geq 0$, thus:

$$
\begin{aligned}
& \mathbb{E}[R_{[y]}(C_i) \wedge R_{[y]}(C_j)] = R_{[y]}(C_i \cup C_j) \\
\iff & |C_i|_y |C_j|_y = 0 \\
\iff & |C_i|_y = 0 \vee |C_j|_y = 0 \\
\iff & R_{[y]}(C_i) = 0 \vee R_{[y]}(C_j) = 0
\end{aligned}
\tag{13}
$$

## A.6 GENERALIZATION TO MULTIPLE CLUSTERS

Let $P_{[y]}(C_j), R_{[y]}(C_j)$ denote the $B^3$ precision and recall scores of label $y$ on elements in a cluster $C_j$. It was shown earlier that Theorems 1 & 2 hold for any two pairwise clusters. Assume now having a fixed but arbitrary number of clusters $C_1, C_2, ..., C_k$, with $k > 2$. The following grouping can now be iteratively applied for $j > 1$:

$$
\begin{aligned}
\boldsymbol{C}_1^* &:= C_1 \\
\boldsymbol{C}_j^* &:= C_j \cup \boldsymbol{C}_{j-1}^*
\end{aligned}
\tag{14}
$$

Notice $\boldsymbol{C}_j^* \triangleq \bigcup_{i<j} C_i$. The above definition allows us to realize that for any tuple pair $(C_j, \boldsymbol{C}_{j-1}^*)$ Theorems 1 & 2 must also hold. Thus, in particular, it holds for the pair $(C_k, \boldsymbol{C}_{k-1}^*)$, and so:

$$\mathbb{E}[P_{[y]}(\bigcup_{j\leq k} C_j)] \leq \mathbb{E}[P_{[y]}(\boldsymbol{C}_{k-1}^*) \wedge P_{[y]}(C_k)]$$

$$\leq \mathbb{E}\big[\mathbb{E}[P_{[y]}(\boldsymbol{C}_{k-2}^*) \wedge P_{[y]}(C_{k-1})] \wedge P_{[y]}(C_k)\big]$$

$$\leq \mathbb{E}\big[\mathbb{E}[...\mathbb{E}[P_{[y]}(\boldsymbol{C}_1^*) \wedge P_{[y]}(C_2)]...] \wedge P_{[y]}(C_{k-1}) \wedge P_{[y]}(C_k)\big] \tag{15}$$

$$\leq \mathbb{E}[P_{[y]}(C_1) \wedge ... \wedge P_{[y]}(C_{k-1}) \wedge P_{[y]}(C_k)]$$

$$\leq \mathbb{E}\big[\bigwedge_{j\leq k} P_{[y]}(C_j)\big]$$

Likewise, it holds that:

$$\mathbb{E}[R_{[y]}(\bigcup_{j\leq k} C_j)] \geq \mathbb{E}[R_{[y]}(\boldsymbol{C}_{k-1}^*) \wedge R_{[y]}(C_k)] \geq \mathbb{E}\big[\bigwedge_{j\leq k} R_{[y]}(C_j)\big] \tag{16}$$

## A.7 Proof of coherence for $\alpha$Max-$B_\delta^3$

Let $C_i, C_j$ be disjoint clusters of distinct elements. Theorem 1 is to be proven for the weighted version: $\mathbb{E}[P_{[y]}^\delta(C_i) \wedge P_{[y]}^\delta(C_j)] \geq \mathbb{E}[P_{[y]}^\delta(C_i \cup C_j)]$, where $P_{[y]}^\delta(C)$ is the ENS-weighted $B^3$ precision scores of label $y$ on elements in $C_j$ for a $\delta_y \in [0,1]$; with weight $w_y := (1-\delta^{|y|})/(1-\delta)$. $|y|$ be the frequency of the class $y$ in the entire data set. Let $|C_i|, |C_j| \in \mathbb{N}$ denote the total number of elements in $C_i, C_j$ and $|C_i|_y, |C_j|_y \in \mathbb{N}$ respectively the number of elements in clusters $C_i, C_j$ with label $y$.

$$\mathbb{E}[P_{[y]}^\delta(C_i) \wedge P_{[y]}^\delta(C_j)] \geq P_{[y]}^\delta(C_i \cup C_j)$$

$$\iff \mathbb{E}\big[w_y^{-1} P_{[y]}(C_i) \wedge w_y^{-1} P_{[y]}(C_j)\big] \geq w_y^{-1} P_{[y]}(C_i \cup C_j)$$

$$\iff w_y^{-1} \frac{|C_i|_y R_{[y]}(C_i) + |C_j|_y R_{[y]}(C_j)}{|C_i|_y + |C_j|_y} \geq w_y^{-1} P_{[y]}(C_i \cup C_j) \tag{17}$$

$$\iff w_y^{-1} \mathbb{E}[P_{[y]}(C_i) \wedge P_{[y]}(C_j)] \geq w_y^{-1} P_{[y]}(C_i \cup C_j)$$

$$\iff \mathbb{E}[P_{[y]}(C_i) \wedge P_{[y]}(C_j)] \geq P_{[y]}(C_i \cup C_j) \overset{Proof A.1}{\iff} \top$$

Since the weight $w_y^{-1}$ cancels out, the proof for Proposition 1 follows immediately. Similarly, it can be shown that Theorem 2 holds as well for $\mathbb{E}[R_{[y]}^\delta(C_i) \wedge R_{[y]}^\delta(C_j)] \leq \mathbb{E}[R_{[y]}^\delta(C_i \cup C_j)]$:

$$\mathbb{E}[R_{[y]}^\delta(C_i) \wedge R_{[y]}^\delta(C_j)] \leq R_{[y]}^\delta(C_i \cup C_j)$$

$$\iff \mathbb{E}\big[w_y^{-1} R_{[y]}(C_i) + w_y^{-1} R_{[y]}(C_j)\big] \leq w_y^{-1} R_{[y]}(C_i \cup C_j)$$

$$\iff w_y^{-1} \frac{|C_i|_y R_{[y]}(C_i) + |C_j|_y R_{[y]}(C_j)}{|C_i|_y + |C_j|_y} \leq w_y^{-1} R_{[y]}(C_i \cup C_j) \tag{18}$$

$$\iff \mathbb{E}[R_{[y]}(C_i) \wedge R_{[y]}(C_j)] \leq R_{[y]}(C_i \cup C_j) \overset{Proof A.4}{\iff} \top$$

Again, since $w_y^{-1}$ cancels out, the respective proof for Proposition 2 follows directly.

## A.8 Proof of non-monotonicity

Following Equations 5, 7, let $\mathbb{E}[\eta_\alpha^{[j]}]$ be a continuous function over the closed interval $\alpha \in [0,1]$ such that $\lim_{\alpha\to 0} \mathbb{E}[\eta_\alpha^{[j]}] = 0^+$ and $\lim_{\alpha\to 1} \mathbb{E}[\eta_\alpha^{[j]}] = \mathbb{E}[\eta^{[j]}]^+$. We show that $\mathbb{E}[\eta_\alpha^{[j]}]$ cannot be monotonic while approaching its limit from above at both endpoints. Suppose, for the sake of contradiction, that $\mathbb{E}[\eta_\alpha^{[j]}]$ is a monotonic function. Since $\lim_{\alpha\to 1} \mathbb{E}[\eta_\alpha^{[j]}] = \mathbb{E}[\eta^{[j]}]^+$, there exists an $\alpha' \in [0,1]$ such

that $0^+ \leq \mathbb{E}[\eta_{\alpha'}^{[j]}] \leq \mathbb{E}[\eta^{[j]}]^+$. By the Intermediate Value Theorem, there must exist an $\alpha'' \in [0, 1]$ such that $\mathbb{E}[\eta_{\alpha''}^{[j]}] = \mathbb{E}[\eta^{[j]}]$. However, this contradicts the assumption that $\mathbb{E}[\eta_{\alpha}^{[j]}]$ approaches its limit from above at both endpoints. Therefore, it cannot be monotonic.

### A.9 THEOREMS' INEQUALITY IMPLICATIONS

Let $\mathbb{S}_i$ be a set of clusters $C_j \subset S_i$ such that $S_i$ is the union of all the elements in the clusters, $S_i := \bigcup_{C_j \in \mathbb{S}_i} C_j$. We show that Theorems 1, 2 imply that $\mathbb{E}_{C_j \in \mathbb{S}_i}[\frac{P_{[y]}(C_j)}{P_{[y]}(S_i)}] \geq 1$; and $\frac{R_{[y]}(C_j)}{R_{[y]}(S_i)} \leq 1$. Let $|S_i|, |C_j| \in \mathbb{N}$ denote the number of elements and $|S_i|_y, |C_j|_y \in \mathbb{N}$ respectively the number of elements with label $y$. $|S_i| \geq |C_j|$ and $|S_i|_y \geq |C_j|_y$ is guaranteed since $C_j \subset S_i$. $|y| \in \mathbb{N}$ is the total number of elements with label $y$ in the entire data set.

Starting with *recall*, where:

$$\frac{R_{[y]}(C_j)}{R_{[y]}(S_i)} = \frac{|C_j|_y}{|y|} / \frac{|S_i|_y}{|y|} = \frac{|C_j|_y}{|S_i|_y} \leq 1. \tag{19}$$

holds trivially per definition since $|S_i|_y \geq |C_j|_y$.

For *precision*, we know:

$$\mathbb{E}_{C_j \in \mathbb{S}_i}\left[\frac{P_{[y]}(C_j)}{P_{[y]}(S_i)}\right] \triangleq \frac{1}{|\mathbb{S}|} \sum_{C_j \in \mathbb{S}_i} \frac{P_{[y]}(C_j)}{P_{[y]}(S_i)} \tag{20}$$

and therefore,

$$\mathbb{E}_{C_j \in \mathbb{S}_i}\left[\frac{P_{[y]}(C_j)}{P_{[y]}(S_i)}\right] \geq 1 \iff \frac{1}{|\mathbb{S}|} \sum_{C_j \in \mathbb{S}_i} \frac{P_{[y]}(C_j)}{P_{[y]}(S_i)} \geq 1$$
$$\iff \frac{1}{|\mathbb{S}|} \sum_{C_j \in \mathbb{S}_i} P_{[y]}(C_j) \geq P_{[y]}(S_i) \iff \frac{1}{|\mathbb{S}|} \sum_{C_j \in \mathbb{S}_i} \frac{|C_j|_y}{|C_j|} \geq \frac{|S_i|_y}{|S_i|} \tag{21}$$

We can re-write the expression $\frac{|S_i|_y}{|S_i|}$ as:

$$\frac{|S_i|_y}{|S_i|} = \frac{\sum_{C_j \in \mathbb{S}_i} |C_j|_y}{\sum_{C_j \in \mathbb{S}_i} |C_j|} = \sum_{C_j \in \mathbb{S}_i} \frac{|C_j|_y}{\sum_{C_k \in \mathbb{S}_i} |C_k|} \tag{22}$$

and since $\sum_{C_k \in \mathbb{S}_i} |C_k| \geq \mathbb{E}[|C_j|] = \frac{1}{|\mathbb{S}|} \sum_{C_k \in \mathbb{S}_i} |C_k|$, we obtain the inequality of:

$$\sum_{C_j \in \mathbb{S}_i} \frac{|C_j|_y}{\sum_{C_k \in \mathbb{S}_i} |C_k|} \leq \sum_{C_j \in \mathbb{S}_i} \frac{|C_j|_y}{\frac{1}{|\mathbb{S}|} \sum_{C_k \in \mathbb{S}_i} |C_k|} \tag{23}$$

and so, with $\frac{1}{|\mathbb{S}|} \sum_{C_k \in \mathbb{S}_i} |C_k| \geq \frac{1}{|\mathbb{S}|} |C_j|$ for all $j$, we have:

$$\frac{1}{|\mathbb{S}|} \sum_{C_j \in \mathbb{S}_i} \frac{|C_j|_y}{|C_j|} \geq \sum_{C_j \in \mathbb{S}_i} \frac{|C_j|_y}{\frac{1}{|\mathbb{S}|} \sum_{C_k \in \mathbb{S}_i} |C_k|} \geq \frac{|S_i|_y}{|S_i|} \tag{24}$$

which proofs $\mathbb{E}_{C_j \in \mathbb{S}_i}\left[\frac{P_{[y]}(S_i)}{P_{[y]}(C_j)}\right] \leq 1$.

## A.10    PROOF OF $\alpha$-PLATEAU

We prove the existence of a function plateau in Equation 7 by proving that the function:

$$f(\alpha) := \frac{\min[p, \alpha q] \min[p_2, \alpha q_2]}{\max[p, \alpha q] \max[p_2, \alpha q_2]} \tag{25}$$

has a plateau $f(\alpha_1) = f(\alpha_2)$ within $\min[a, a_2] \leq \alpha_1 \leq \alpha_2 \leq \max[a, a_2]$ for $\alpha_1, \alpha_2 \in [0, 1]$; given $\nu_1 = \frac{p_1}{q_1}$, $\nu_2 = \frac{p_2}{q_2}$, and fixed $p_1, q_1, p_2, q_2$. First, we consider the fact that:

$$g_{p,q}(\alpha) := \frac{\min[p, \alpha q]}{\max[p, \alpha q]} = \begin{cases} \frac{\alpha q}{p} & \text{if } \alpha \leq \frac{p}{q} \\ \frac{p}{\alpha q} = (\frac{\alpha q}{p})^{-1} & \text{otherwise} \end{cases} \tag{26}$$

which holds since $\min[p, \alpha q] = \alpha q \iff \alpha q \leq p \iff \alpha \leq \frac{p}{q}$, and $\max[p, \alpha q] = -\min[-p, -\alpha q]$. W.l.o.g, assume $\nu_1 \leq \nu_2$, and let $\alpha \in [\nu_1, \nu_2]$. Since $\alpha_1 \geq \nu_1$, it holds $g_{p_1, q_1}(\alpha) = (\frac{\alpha q_1}{p_1})^{-1}$. Similarly, $\alpha \leq \nu_2 \implies g_{p_2, q_2}(\alpha) = \frac{\alpha q_2}{p_2}$. Therefore, $\nu, \nu_2$ are the two inversion points:

$$f(\alpha) = g_{p_1, q_1}(\alpha) \cdot g_{p_2, q_2}(\alpha) = \frac{p_1}{\alpha q_1} \frac{\alpha q_2}{p_2} = \frac{p_1 q_2}{q_1 p_2} = \nu_1 \nu_2 \tag{27}$$

which is independent of $\alpha$ within that interval and, thus, *constant* (i.e. a plateau).

