# OpenReview forum: "αMax-B-CUBED: A Supervised Metric for Addressing Completeness and Uncertainty in Cluster Evaluation"
_ICLR.cc/2024/Conference — Submitted to ICLR 2024_

### Official Review · Reviewer_mjBX · 2023-10-27

**Soundness:** 2 fair
**Presentation:** 1 poor
**Contribution:** 2 fair
**Rating:** 3
**Confidence:** 5

**Summary:**

The paper proposes an extension of a supervised (external) cluster evaluation measure called B-CUBED (B^3). The extension, called aMax-B^3, incoropates a parameter a specifying the uncertainty related to the existence of sub-clusters within the clusters of a ground truth class.
It is also claimed that the proposed measure is more robust and fair in the case of imbalanced datasets.

**Strengths:**

-The proposed measure seems to be a novel extension of B^3 measure for supervised cluster evaluation.
-The paper includes some theoretical proofs.

**Weaknesses:**

- The paper lacks considerably in terms of presentation and clarity (see questions below).
- B^3 is not a widely used measure in the clustering literature (such as NMI for example).
- The experimental part is weak and does not involve real datasets.
- The proposed measure could have been compared not only with B^3 but also with other measures (e.g. NMI).

**Questions:**

1) Presentation of the essential part of the approach in pages 5 and 6 is poor and hard to follow. There are several incomplete sentences and the use of indices i, j and k causes a lot of confusion. For example in eq. (5), \eta^[j]=|C_j|/|S_i| seems to depend also on i.
2) Before section 4.1 it is mentioned that "the final F_\beta score is", but F_b is not presented afterwards.
3) It is not clear how \alpha is computed or estimated. This is a major issue in the paper.
4) The results in Figure 2 and 3 need a much better explanation.
5) In the plot of Figure 3 the x-axis corresponds to number of clusters, while in the legend it is mentioned that corresponds to cluster size.
6) Experiments with real datasets would add value to the paper. Also it would be interesting to show how other measures such as NMI compare to the proposed measure.

---

> ### Author Response · Authors · 2023-11-13
>
> We have updated the draft based on author's feedback.
>
> ---
>
> **Feedback:**
>
> $B^3$ is not a widely used measure in the clustering literature (such as NMI for example).
>
> **Response:**
>
>  In the clustering literature, $B^3$ is not as commonly employed as metrics like NMI (Normalized Mutual Information).
> However, when it comes to supervised evaluation, B^3 is the only measure that satisfies multiple formal constraints.
> The paper titled "A comparison of extrinsic clustering evaluation metrics based on formal constraints" has gained significant recognition with over 1022 citations, highlighting its relevance, and hence can be considered common for supervised settings.
> Mutual information has also be shown to not satisfy all constraints, unlike B3 paper for details and proof in original paper.
>
> ---
>
> **Feedback:**
>
> The proposed measure could have been compared not only with B^3 but also with other measures (e.g. NMI).
>
> **Response:**
>
> NMI is an unsupervised metric. We mainly focus on supervised metrics instead of unsupervised metrics like NMI. This decision is based on the discussion in the original B^3 paper that highlights the necessity and advantages of supervised evaluation. Our work specifically addresses the "completeness" constraint within the supervised metric framework, particularly for the B^3 metric. By directly comparing our approach to B^3, we aim to provide a comprehensive evaluation that highlights the problem. It would not be logical to compare our method against other metrics because our goal is not to assert the superiority of B^3 over other algorithms. Instead, we aim to demonstrate how the completeness problem, when constrained within the supervised B^3 setting, can degrade the quality of results. To solve this issue, we propose a direct mathematically proven solution.
>
> ---
>
> **Feedback:**
>
> Presentation of the essential part of the approach in pages 5 and 6 is poor and hard to follow. There are several incomplete sentences and the use of indices i, j, and k causes a lot of confusion. For example in eq. (5), $\eta^{[j]}=|C_j|/|S_i|$ seems to depend also on i.
>
> **Response:**
>
> We have revised the text to offer a more comprehensive, intuitive, and concise definition. In particular, indices "i" and "j" are now clear. Please refer to the updated PDF submission for the LaTex formatted version:
>
> The weight $\eta^{[j]}$ assigned to cluster $C_j$ is calculated as the ratio of the number of elements in $C_j$ to the total number of elements in the super-set $S_i$ to which it is assigned. Although $\eta^{[j]}$ depends on $i$ since every $j$ is uniquely identified by a super-set $i$, the use of $j$ suffices.
>
> Let $\mathbb{S}$ be the collection of all clusters. A super-set $S_i \subset \mathbb{S}$ is the union of a specific list of clusters indexed by $C_j$, i.e., $S_i := \bigcup_{j\in \mathcal{J}} C_j$, where $\mathcal{J}$ represents the set of cluster indices. Each $C_j$ is always assigned to exactly one $S_i$. The set of all identified super-clusters is denoted by $\hat{S}$. Elements in $\hat{S}$ are mutually disjoint. The number of elements in a collection is denoted by $|S_i|$, $|C_j|$, and $|S_i|_y$, $|C_j|_y$ is the number of elements with label $y$ in the respective collection, where $|y|$ indicates the total number of all elements with label $y$.
>
> ---
>
> **Feedback:**
>
>  Before section 4.1 it is mentioned that "the final F_\beta score is," but F_b is not presented afterwards.
>
> **Response:**
>
> We have provided the scores for the alpha-precision and alpha-recall versions of the original beta score. The final score of B^3 is simply the F_\beta score based on precision and recall. The definition of the final alpha-b^3 score is theoretically implied by this, but we have updated the PDF to explicitly mention it.
>
> ---
>
> **Feedback:**
>
> It is not clear how \alpha is computed or estimated. This is a major issue in the paper.
>
> **Response:**
>
> We wrote before section 4.1: "Although $\alpha$ can be set manually, we assume no prior knowledge on label uncertainty and set $\alpha:= \min(p_1,p_2)$ as the default choice for extracting the highest possible $\eta^{[j]}_\alpha$ weight with minimal uncertainty."
> We consider the automatic determination of alpha to be crucial, and have attempted to provide a mathematically precise definition.
>
> ---
>
> **Feedback:**
>
> The results in Figure 2 and 3 need a much better explanation.
>
> **Response:**
>
> We visualized the B3 scores in connection with the determined clusters to provide insight into how they relate to the visual clustering split for various cluster sizes. The results can be understood in terms of how both the original B3 score and the alpha-max B3 score would assess them. We have updated the caption for a better explanation (see updated PDF);.
>
> ---
>
> **Feedback:**
>
> The results in Figure 2 and 3 need a much better explanation.
>
> **Response:**
>
> We corrected this ambiguity by renaming the axes more appropriately.

---

> > ### Comment · Reviewer_mjBX · 2023-11-22
> > **Acknowledgement**
> >
> > I thank the authors for the reply. I have read their responses and the updated pdf.

---

### Official Review · Reviewer_KXAB · 2023-10-27

**Soundness:** 3 good
**Presentation:** 2 fair
**Contribution:** 2 fair
**Rating:** 5
**Confidence:** 4

**Summary:**

In this paper, a Max-algorithm is proposed to solve the problems of ambiguity and uncertainty in labels. Max-algorithm considers the completeness and uncertainty of subgroup evaluation. The results show that the method is suitable for subgroup uncertainty in basic labels and can be extended to unbalanced data sets. Compared to technology, the Max-algorithm can produce more robust and fair results and adapt to label uncertainties.

**Strengths:**

1. The proposed method is an extension of the clustering evaluation index and has a positive effect in this field.
2. The proposed method is sound technically.
3. The theoretical foundation seems to be relatively sufficient.

**Weaknesses:**

1. This work is not innovative enough and the writing storyline is average.
2. The comparison algorithms used in the experimental part are few, only one has been mentioned. And the comparison experiments with more clustering measures should be added.
3. The results of this paper are not presented well, and Figure 3 is very rough.

**Questions:**

1. In this paper, all other theorems and corollaries are proved, but Corollary 1 is not.
2. The only comparative evaluation index is B3. Are there no other similar evaluation indexes?
3. In Figure 3, only the left picture is introduced, and the right subgraphs are not introduced for specific. It is not clear whether this subgraph is the result of B3 or M-B3. After all, in Figure 2 maxB3 is not marked with a yellow circle. It is best to unify the format of all clustering subgraphs. Also, in the top left plot of Figure 3, B3 looks like there is no result in a cluster number of 1-4. It turns out that the results overlap. You need to re-adjust the color of the image to make the results more obvious.
5. In the last paragraph of the introduction, it is recommended that the author explain the effectiveness of the proposed method.
6. Should the uncertainty of labels be tested with different types of noise? It is recommended that the author increase the type of label noise to make the method proposed in this article more convincing.

---

> ### Author Response · Authors · 2023-11-13
>
> Updated draft based on author's feedback
>
> ---
> **Feedback:** The comparison algorithms used in the experimental part are few, only one has been mentioned, and the comparison experiments with more clustering measures should be added.
>
> **Response:** Our focus is on addressing the completeness flaw in the supervised B^3 evaluation framework, and we compare our proposed method with the traditional B^3 metric. Our goal is not to assess the superiority of B^3 over other algorithms, and we believe that solely concentrating on comparing against B^3 without involving other metrics best meets our works' intention. Additionally, comparing clustering algorithms is non-trivial due to fairness as each algorithm performs differently well on different data sets, and we don't include other metrics for this reason.
>
> ---
>
> **Feedback:** The results of this paper are not presented well, and Figure 3 is very rough.
>
> **Response:** We have updated the PDF with an explanation of Figure 3. In the updated draft, we state that we evaluated multiple clustering assignments through clustering scores and visual illustrations on an imbalanced dataset with five labels, where the ideal clustering consists of five classes. Finer and purer subclustering achieves better scores compared to coarse and less pure clustering.
>
> ---
>
> **Feedback:** In this paper, all other theorems and corollaries are proved, but Corollary 1 is not.
>
> **Response:** We have updated our text, and Corollary 1 is now mentioned explicitly. We had omitted the corollary, assuming that it could be evident or directly following from Proposition 1 and Theorem 1. We have now included the proof of Corollary 1 in the appendix.
>
> ---
>
> **Feedback:** The only comparative evaluation index is B3. Are there no other similar evaluation indexes?
>
> **Response:** We propose a mathematically devised solution to the completeness flaw in the supervised B^3 evaluation framework, addressing its limitations. Our goal is not to assess the superiority of B^3 over other algorithms, and we focus on comparing our proposed method with B^3 only. Additionally, comparing clustering algorithms is complex and challenging due to fairness as each algorithm performs differently on different datasets.
>
> ---
>
> **Feedback:** In Figure 3, only the left picture is introduced, and the right subgraphs are not introduced for specific. It is not clear whether this subgraph is the result of B3 or M-B3.
>
> **Response:** We have updated Figure 3 in the new PDF, and we clarify that the graphs show the $B^3$, $\alpha$Max-$B^3$, and $\alpha$Max-$B^3_\delta$ scores on an imbalanced dataset with five labels. These scores are evaluated based on different cluster counts $k.$ On coarse clusters ( $k \leq 5$), $B^3$ and $\alpha$Max-$B^3$ perform equally. However, $\alpha$Max-$B^3$ performs better on finer sub-clusters, giving more weight to sub-clusters with uncertainty. $\alpha$Max-$B^3_\delta$ accounts for class imbalance, resulting in differences in $k<5.$
>
> ---
> **Feedback:** In the last paragraph of the introduction, it is recommended that the author explain the effectiveness of the proposed method.
>
> **Response:** We have updated the last paragraph of the introduction in the new PDF to better explain the effectiveness of our proposed method. We state that the traditional B^3 metric may not provide accurate evaluation for clustering outcomes on finer subgroups or coarse labels, and we address this limitation by suggesting a modified mathematical formula for B^3; with the formula incorporating a super-aggregation of the cluster groups into its scoring function and aiming to improve the evaluation process's quality.
>
> ---
>
> **Feedback:** Should the uncertainty of labels be tested with different types of noise? It is recommended that the author increase the type of label noise to make the method proposed in this article more convincing.
>
> **Response:** Our focus is on addressing the completeness flaw in the supervised B^3 evaluation framework. While we consider uncertainty based on the completeness flaw, the noise we're referring to isn't random noise. Instead, it refers to label uncertainty when using broad categories. alpha-BCubed addresses this uncertainty, uncertainty != noise.  E.g., we have a dataset of images labelled as Fruit or Vehicles, and we want to evaluate a deep learning model's performance by using the B^3 metric on embeddings alone. Consider a model A that clusters all fruits and vehicles into separate groups, receiving a perfect score according to completeness constraint and B^3 principles. However, model B produces multiple sub-clusters, grouping similar fruits (apples, bananases) and vehicles (planes, cars, etc.) together. Although this results in many correct sub-clusters, the evaluation is worse than the original B^3. Yet, we're more interested in the second model, and the original B^3 would provide a sub-optimal choice for model decision. Hence , in alpha-BCubed the alpha value denotes uncertainty rather than noise.

---

### Official Review · Reviewer_AXE2 · 2023-10-31

**Soundness:** 2 fair
**Presentation:** 3 good
**Contribution:** 3 good
**Rating:** 3
**Confidence:** 4

**Summary:**

The extrinsic B-CUBED metric (precision, recall, and the F-score) is one of the most common clustering evaluation metric. However, this metric does not work well for unbalanced datasets and implicitly assumes that the labels are correct and there are no (relevant) sub-clusters inside groups of equally labeled objects. To address the above issues, the author provides a more fair evaluation metric that is applicable to unbalanced datasets and datasets with uncertain labels. The effectiveness of the proposed metric was verified by clustering experiments on artificial datasets.

**Strengths:**

The authors provide a new extrinsic clustering evaluation metric that can be applied to unbalanced datasets and labeled uncertain datasets.

**Weaknesses:**

The authors provide a new extrinsic evaluation metric for clustering methods that is innovative. However, the paper evaluates the proposed metric using clustering results of k-means for a special artificial dataset, and the results only show higher values compared to the existing B-CUBED metric, and do not demonstrate the advantages of the proposed metrics. A good metric should be able to discover the true structure of the data more accurately in real data experiments compared to existing metrics, and the paper's experiments do not verify this point. Meanwhile, the explanation of notation on the key formula (8) is not clear, leading to difficulties in understanding the evaluation metrics. There are the following minor problems:
(1)	The references of the paper are too old and lack research on the latest work.
(2)	There are some minor errors in the paper, please check carefully, such as in Proposition 1, the formula is missing half a bracket.

**Questions:**

The author should experimentally verify that the proposed evaluation metric has obvious advantages compared with existing evaluation metrics. If two metrics have a positive correlation, for example, both are large and both are small, it does not indicate the advantage of the proposed metric.

---

> ### Author Response · Authors · 2023-11-13
>
> We have updated the Draft based on author's feedback.
>
> **Feedback:**
>
> The author should experimentally verify that the proposed evaluation metric has obvious advantages compared with existing evaluation metrics. If two metrics have a positive correlation, for example, both are large and both are small, it does not indicate the advantage of the proposed metric.
>
> ...and the results only show higher values compared to the existing B-CUBED metric, and do not demonstrate the advantages of the proposed metrics.
>
> **Response:**
>
> Our study focuses primarily on supervised metrics, with particular emphasis on the $B^3$ metric. This metric has gained widespread recognition and has been extensively cited in the literature. It is considered a common measure for supervised evaluation due to its ability to satisfy multiple formal constraints.
>
> The decision to concentrate on supervised metrics, as opposed to unsupervised metrics, stems from the advantages and necessity highlighted in the original $B^3$ paper. In our work, we specifically address the "completeness" constraint within the supervised metric framework, focusing on refining the well known $B^3$ metric. By directly comparing our approach with $B^3$, we aim to provide a comprehensive evaluation that sheds light on the associated problem.
>
> It is important to note that our intention in this work is not to establish the superiority of $B^3$ over other algorithms, or advocating for $B^3$. In clustering, different algorithms perform differently well on different data sets, and there is no silver bullet clustering algorithm that outperforms all other ones. Each has its advantages and disadvantages, and fair comparisons are not always possible. Rather, our primary objective is to demonstrate the negative impact of the completeness problem within the well known supervised $B^3$ metric on the quality of results. Thus, we propose a mathematically proven solution to address this issue directly and use experiments to support our theoretical claims.
>
> Considering the prominence and relevance of the $B^3$ metric within the field of supervised evaluation, alongside the constraints it satisfies that are not met by other metrics, our focus remains on improving and refining this specific evaluation framework.
>
> Let's consider two scenarios:
> (1) In the first scenario, a model clusters all fruits together and all vehicles together. According to the completeness constraint and $B^3$ principles, this arrangement would receive a perfect score. Other unsupervised metrics could not be used in this context, because we are in a supervised evaluation setting. (2) Now, in the second scenario, the model not only separates fruits and vehicles accurately but also creates additional subclusters within each category. For example, it groups all apples, bananas, and coconuts together as subclusters, and all planes, cars, trains, and ships as another subcluster. Although this results in multiple correct subclusters, the evaluation metric would be worse than the original $B^3$ metric. However, our primary interest lies in the performance of the second model, as it successfully identifies similarities within each category. Using the standard $B^3$ metric in this case would be a suboptimal choice for making model decisions.
>
> **Feedback:** There are some minor errors in the paper, please check carefully, such as in Proposition 1, the formula is missing half a bracket.
>
> **Response:** Indeed. We have corrected some formula mistakes in the newest pdf version. We have further added better explanations to the figures. We have also rewritten the notation better above Formula 5, and adjusted notation above Formula 8 slightly.

---

### Official Review · Reviewer_6ExU · 2023-11-03

**Soundness:** 4 excellent
**Presentation:** 3 good
**Contribution:** 3 good
**Rating:** 8
**Confidence:** 2

**Summary:**

The paper presents a metric for clustering evaluation based on the earlier B-cube clustering evaluation metric. The new metric addresses a weakness in evaluating the completeness constraint. The B-cube metric favours larger clusters although practically, an algorithm making smaller size clusters may be preferred. The new metric also accounts for imbalanced data sets. By setting the value of uncertainty, it can be controlled whether sub-groups of a cluster are required or not. If not, the measure gives the same results as b-cube

**Strengths:**

The paper is well written, and the proposed measure has a sound mathematical background. The authors have clearly described the case where the original metric may be problematic and have thus built a case for their metric

**Weaknesses:**

The paper contributes by suggesting an improvement in the original metric. The authors provide a sound background for their work. However, it is not clear how significant this improvement is practically, since they have used a very small set of clusters as the ground truth, and a very short experimental results section.

**Questions:**

1) Sections 5.1 & 5.2 could have provided more detail about the results.

2) A very small sized ground truth dataset with 5 clusters has been used. The authors state that when k<=5, b-cube and the new proposed metric give the same results. Why wasn't a larger dataset used

---

> ### Author Response · Authors · 2023-11-13
>
> We have updated the Draft based on author's feedback.
>
> ------------------------------------------------------------------------------------------------------------
>
> **Feedback:**
>
> It is not clear how significant this improvement is practically, since they have used a very small set of clusters as the ground truth, and a very short experimental results section.
>
> **Response:**
>
> Our study focuses on refining the $B^3$ metric, which has become a widely recognized and extensively cited supervised evaluation measure. We address the completeness constraint within this metric and propose a mathematically supported motivation to improve the quality of results. A key message we believe to contribute by is in particular that to demonstrate the negative impact of the completeness problem on evaluation in the original $B^3$ metric.
>
> Our motivations comes form the following observation / use-case, where given two scenarios, the standard $B^3$ metric may be suboptimal for decision-making in clustering models. For example: [case 1]: Imagine a model that clusters all fruits together and all vehicles together. In this supervised evaluation setting, other unsupervised metrics cannot be utilized. According to the completeness constraint and the principles of the $B^3$ metric, this would receive a perfect score. [Case 2]: Imagine now a second model that not only correctly separates fruits and vehicles but also creates additional subclusters within each category. For instance, it groups apples, and coconuts together as subclusters, and planes and ships as another subcluster. Although this results in multiple correct subclusters, the standard $B^3$ evaluation metric would be worse than using the original $B^3$ metric. However, the focus here is on the performance of the second model in comparison two the first model, not to other evaluation metrics.
>
> Our work intends to highlight the flaw of using the standard $B^3$ metric in relation to the completeness theorem, as it may not accurately reflect the model's ability to identify nuanced subclusters within categories. In other words, we want to demonstrate that the standard $B^3$ metric may not always be the optimal choice for evaluating clustering models when the objective extends beyond purely categorizing distinct groups.
>
> We consider the focus of our is on refining the $B^3$ metric and addressing the completeness problem within a supervised evaluation framework.
>
> ------------------------------------------------------------------------------------------------------------
>
> **Feedback:**
>
>  Sections 5.1 & 5.2 could have provided more detail about the results.
>
> **Response:**
>
> We have updated and improved the captions and explanations of the results, and figures (see updated PDF version).
>
> ------------------------------------------------------------------------------------------------------------
>
> **Feedback:**
>
> A very small sized ground truth dataset with 5 clusters has been used. The authors state that when $k \leq 5$, $b$-cube and the new proposed metric give the same results. Why wasn't a larger dataset used?
>
> **Response:**
>
> Here we align partly with our previous response. One argument are limitations and difficulties in visualizing clusters when there are numerous clusters and complex splits. E.g. visualizing more than 5 clusters and multiple splits can cause confusion and hinder the clear illustration of cluster formation and definition. However, a key result of the study is the accurate prediction of uncertainty alpha for balanced data and the approximation for unbalanced data. This outcome we considered a stronger and more significant result in our research. In general, page limitations imposed doesn't allow us to include all experiments. We believed the inclusion of a graphical representation and experimental results on uncertainty estimation were important.

---

### Comment · Area_Chair_gCrx · 2023-11-22
**Author-Reviewer Discussion ends soon**

Dear Reviewers and Authors,

The discussion phase ends soon. Please check all the comments, questions, and responses and react appropriately.

This is extremely important for this paper as it has received very extreme ratings.

Thank you!

Best, AC for Paper #241

---

### Meta-Review · Area_Chair_gCrx · 2023-12-11

**Metareview:**

The authors consider the problem of measuring the quality of clustering with "supervised" metrics. They generalize the well-known B3 measures by weakening the so-called completeness constraints. Unfortunately, the paper in the current state is not mature enough, with rather a narrow scope, and limited experiments that do not contain any real-world use cases.

The authors were trying to defend their work, but even the most positive reviewer wrote that "it is not clear how significant this improvement is practically, since they have used a very small set of clusters as the ground truth, and a very short experimental results section."

**Justification For Why Not Higher Score:**

The paper is not mature enough, with rather a narrow scope, and without any real-world use cases. Also the criticism of the completeness constraints is not clearly presented and rather debatable.

**Justification For Why Not Lower Score:**

N/A

---

### Decision · Program_Chairs · 2024-01-16

Reject